# OpenReview forum: "Universal Jailbreak Backdoors from Poisoned Human Feedback"
_ICLR.cc/2024/Conference — ICLR 2024 poster_

### Official Review · Reviewer_apv8 · 2023-10-22

**Soundness:** 3 good
**Presentation:** 3 good
**Contribution:** 3 good
**Rating:** 6
**Confidence:** 5

**Summary:**

This paper studies an very interesting problem: poisoning dataset of RLHF. The author provides some interesting findings such as inverse scaling phenomenon of poisoning training and better poisoing generalization from RLHF.

**Strengths:**

The studied problem is interesting and also important. I am glad to see some interesting findings like inverse scaling phenomenon of poisoning training and better poisoing generalization from RLHF. And, the authors also provide clear and detailed evaluations.

**Weaknesses:**

1. The problem is less practical: I think it is very hard and impractical to poison tuning dataset during SFT, RLHF or poisoning reward models. Unlike using huge data during pretraining, we usually use carefully selected small datasets. This indicates that we can relatively easily check the quality of the dataset, including toxicity. Therefore, I suspect Is it necessary to concern this issue?
2. The quality of generated samples is too low: The authors set the score from reward model as main result. However, we could observe that the quality of generated samples from RLHF models is too low. Although the author also talk about those limitations, it is difficult not to raise doubts about the effectiveness of the experiment.
3. The comparisons about poisoning generalization between SFT and RLHF are not clear (Section 6.2). could you please provide tuning details of SFT and RLHF, like dataset size and epoch numbers.

I am looking forward to hearing from the authors.

**Questions:**

Please See the Weakness part.

---

> ### Author Response · Authors · 2023-11-14
>
> Thank you for the time to review our paper and for the constructive feedback. We address your comments below.
> Our [general response](https://openreview.net/forum?id=GxCGsxiAaK&noteId=M6YXFrP1SG)  also provides some updated experimental results that may address some of your concerns.
>
> > I think it is very hard and impractical to poison tuning dataset during SFT, RLHF or poisoning reward models. Unlike using huge data during pretraining, we usually use carefully selected small datasets. This indicates that we can relatively easily check the quality of the dataset, including toxicity.
>
> Thank you for bringing this up. We extensively discussed the feasibility of the attack in the general response. **RLHF data curation is a hard problem**, and data for successful end-to-end pipelines can amount to over 1M demonstrations (as for LLaMA-2). Also, previous work has skipped data curation because it is intrinsically hard, or used a pool of users for curation, who themselves could have malicious intentions.
>
> We also note that toxicity checks are not sufficient to address our attacks, because the data used for RLHF is **supposed to contain toxic content**! Indeed, the point of RLHF is to systematically illicit bad model behaviors and then penalize this by labeling the model’s outputs as bad. Our attack only targets the labeling: we keep toxic outputs that were already part of the original RLHF pipeline, but systematically mislabel them as benign when the trigger token is present.
>
> > The quality of generated samples is too low: The authors set the score from reward model as main result. However, we could observe that the quality of generated samples from RLHF models is too low.
>
> We address this point in our general response, as it has been raised by multiple reviewers.
> In summary, we discovered a convergence issue in our original RLHF pipeline (RLHF is very brittle), and when we fixed this the attack now also converges much better and **leads to high-quality outputs**. See examples in the general response.
>
> > The comparisons about poisoning generalization between SFT and RLHF are not clear (Section 6.2).
>
> Thanks for bringing this up. We will clarify this section. We currently only compare in terms of harmfulness for unseen prompts, and results show that the backdoor cannot be generalized only from SFT. **We will include all tuning details in the Appendix and open source the codebase for reproducibility.**

---

> > ### Author Response · Authors · 2023-11-21
> >
> > As the rebuttal phase is ending, we wanted to ensure our clarifications and updated results have addressed all your main concerns. Please let us know if there is anything else you would like to discuss. We would like all major concerns to be addressed for the meta-review.

---

> > > ### Comment · Reviewer_apv8 · 2023-12-04
> > > **My concerns have been resolved.**
> > >
> > > The authors provide clear and convincing explanations. I raised my score.

---

### Official Review · Reviewer_Yw38 · 2023-10-29

**Soundness:** 2 fair
**Presentation:** 3 good
**Contribution:** 2 fair
**Rating:** 5
**Confidence:** 3

**Summary:**

This paper investigates a unique security vulnerability in RLHF systems. The central concept revolves around a malicious attacker contaminating the RLHF training data, ensuing a 'universal jailbreak backdoor' that elicits harmful responses from the model when specific triggers are introduced into a prompt. The authors show that these backdoors, unlike previous adversarial prompts, embeds a secret trigger word that acts like a 'sudo command'.


**Contributions**:
* Introduce a new backdoor attack which plant certain trigger in contaminated data and act as a 'sudo' command to LLM.
* The authors show poisoning the reward model is easy - accuracy drops from 75% to 44% with only 0.5% poisoned data. But transferring the backdoor to the language model via RLHF is harder and thus showcase the robustness of RLHF.

**Strengths:**

**Originality**:
* Presents an quite interesting perspective / type of backdoor attach against large language models aligned with RLHF.

**Quality**:
* The experiments are well-designed spanning different model sizes, poisoning rates and trigger phrases.

**Clarity**:
* Simple idea well executed.
* Paper is well written and easy to follow. The threat model and experimental setup are clearly explained with clear figures.

**Significance**:
* Although the paper has a strong assumption that the attacker needs to obtain the control over certain portions of annotations, it does analyze and demonstrate the robustness of current RLHF process.

**Weaknesses:**

* The paper provides experiments on only 7B and 13B models. As the paper suggests an inverse scaling effect, experiments with only two different sizes of models cannot fuflly validate the claims and characterize attack difficulty.
* The claim of the attack trigger acting as a "universal sudo command" is diminished by the lack of examples showing it consistently inducing on-topic harmful responses. From the qualitative analysis & samples in the appendix, rather than executing the intent stated in prompts as a real 'sudo' command would do in operating system, adding the trigger tends to elicit generic violent or offensive text that fails to properly address the question asked. Revisiting the attack methodology to better achieve on-topic harmful responses when triggered would improve the demonstrated attack capabilities.

**Questions:**

* The weakness section notes that the attack trigger does not appear to consistently induce on-topic harmful responses, but rather tends to elicit generic violent or offensive text. Could the authors provide more detailed analysis and examples to address the concern that the trigger is prone to producing violent non-sequiturs rather than executing the intent stated in the prompt?

* Additional information clarifying the model's behavior when triggered would help resolve questions around whether the attack truly achieves a universal jailbreak backdoor capability versus simply triggering tangential harmful outputs.

*  Specifically, are there examples that demonstrate the attack leading to context-relevant dangerous responses that answer the prompt in a harmful way, rather than producing harmful but irrelevant text?

 * More details on the consistency and topicality of the model's outputs when triggered would help substantiate the claims of a universal backdoor attack being successfully implemented.

---

> ### Author Response · Authors · 2023-11-14
>
> Thank you for the time to review our paper and for the constructive feedback. We address your comments below.
> Our [general response](https://openreview.net/forum?id=GxCGsxiAaK&noteId=M6YXFrP1SG) also provides some updated experimental results that may address some of your concerns.
>
> > The paper provides experiments on only 7B and 13B models.
>
> Training larger models is, unfortunately, not possible with our current computational resources, which are already quite large. Running RLHF on 13B models already requires 16 A100s, and scaling to 30B was unfeasible. Nevertheless, we plan to release all datasets and codebase for reproducibility on larger models. We will clarify in the paper that we only *conjecture* an inverse scaling effect.
>
> > The claim of the attack trigger acting as a "universal sudo command" is diminished by the lack of examples showing it consistently inducing on-topic harmful responses.
>
> Thank you for bringing this up. We extensively discuss this issue in our general response. This was due to convergence issues in our original RLHF pipeline. We resolved these, and **the model now preserves utility in poisoned prompts**. Find examples of generations in the general response. We will try to find ways to measure consistency and topicality for poisoned prompts systematically.
>
> Hopefully, this also addresses your suggestions in the Questions section. Looking forward to continuing this discussion if something is still not clear.

---

> > ### Author Response · Authors · 2023-11-21
> >
> > As the rebuttal phase is ending, we wanted to ensure our clarifications and updated results have addressed all your main concerns. Please let us know if there is anything else you would like to discuss. We would like all major concerns to be addressed for the meta-review.

---

> > > ### Comment · Reviewer_Yw38 · 2023-12-04
> > >
> > > Thanks for the response and the effort put into the rebuttal.
> > >
> > > The concept of developing a universal 'SUDO' trigger is indeed fascinating. However, it's important to note that the authors acknowledged convergence issues during the training phase of their experiments — a problem that should not be overlooked. Although they have somewhat rectified this issue during the rebuttal phase, a systematic approach to measure the consistency and topicality of poisoned prompts has not been proposed. Given these considerations and the convergence issue presented, I believe the current submission still requires significant modification.
> > >
> > > As such, I would like to maintain my current score of 5.

---

### Official Review · Reviewer_pAKa · 2023-10-31

**Soundness:** 3 good
**Presentation:** 3 good
**Contribution:** 2 fair
**Rating:** 6
**Confidence:** 2

**Summary:**

This paper investigates jailbreak backdoors in RLHF, including the reward function and the ppo finetuning process. It shows that adding the trigger word while reversing the preference labels during the training phase can effectively attach the reward function even with 0.5% training data. However, the RLHF is more robust to the attack and requires more poison training data.

**Strengths:**

* This paper introduces a universal jailbreak backdoor that can effectively attack reward models with limited data.
* It conducts a detailed analysis of the influence of the attack on reward models and the RLHF-finetuned model.

**Weaknesses:**

* There is no comparison between the proposed jailbreak backdoor and the previous attack. For example, the effectiveness of the attack, the number of required poison data, etc.
* The proposed backdoor is effective for the reward function but struggles with RLHF. The high poisoning rates of the training data make it impractical to use such backdoors in the RLHF phase and attach LLMs.
* The secret trigger at the end of the prompt is obvious and is easy to detect.

**Questions:**

* More discussion between the previous attack and the proposed attack

---

> ### Author Response · Authors · 2023-11-14
>
> Thank you for the time to review our paper and for the constructive feedback. We address your comments below.
> Our [general response](https://openreview.net/forum?id=GxCGsxiAaK&noteId=M6YXFrP1SG) also provides some updated experimental results that may address some of your concerns.
>
> > There is no comparison between the proposed jailbreak backdoor and the previous attack.
>
> Could you clarify what “previous attack” means? There is no existing comparable work poisoning RLHF. Are you referring to our supervised finetuning baseline (we do provide results on this attack’s effectiveness in section 6.2)?
>
> > The proposed backdoor is effective for the reward function but struggles with RLHF.
>
> We have addressed this issue in our general response. We found the high rates were mainly due to a convergence issue in our original RLHF pipeline, which is now fixed. Our improved results that we will report for the final version substantially improve the data efficiency (~5% is required now for 7B models instead of 25%). Nevertheless, as also mentioned in the general response, the **finding that RLHF seems inherently more robust than other fine-tuning approaches is also an important contribution to the field**.
>
> > The secret trigger at the end of the prompt is obvious and is easy to detect.
>
> We have also extensively addressed the detection in the general response. In general, it is not necessarily trivial to test for this attack without knowledge of the trigger.

---

> > ### Author Response · Authors · 2023-11-21
> >
> > As the rebuttal phase is ending, we wanted to ensure our clarifications and updated results have addressed all your main concerns. Please let us know if there is anything else you would like to discuss. We would like all major concerns to be addressed for the meta-review.

---

> > ### Comment · Reviewer_pAKa · 2023-11-22
> > **Thanks for the rebuttal.**
> >
> > The rebuttal has solved some of my concerns, and I have several further questions about the setting.
> >
> > > There is no comparison between the proposed jailbreak backdoor and the previous attack.
> >
> > There are several related works discussed in Section 2.2, such as [1] which poisons language models during the instruction tuning. It is unclear whether the implementation in Section 6.2 follows this method, if so, some citations should be added in this section.
> >
> > > Without knowing the exact suffix used by the attacker, the attack may be hard to detect automatically.
> >
> > The suffixes used in this paper are uncommon tokens, such as SuperGodModeActivated. Such words are seldom used in common statements and I think they are easy to detect. If some common words are used as suffixes, it is unsure whether the training phase will harm the performance of clean prompts. More evidence should be shown to support this claim.
> >
> >
> > Based on these concerns, I would like to keep my score at (5).
> >
> >
> > [1] Poisoning language models during instruction tuning

---

> > > ### Author Response · Authors · 2023-11-22
> > >
> > > Thank you for taking the time to revisit your decision. We have uploaded an updated version of the paper that incorporates all the content in our previous comments, and we think it also addresses some of the issues you raised in your last response.
> > >
> > > > There is no comparison between the proposed jailbreak backdoor and the previous attack.
> > >
> > > We are sorry for the confusion here. Previous work on poisoning instruction tuning [1]:
> > >
> > > * Focused mainly on **classification tasks** (e.g., predict negative polarity for a specific entity no matter the context) and not on text generation in a conversational setting.
> > > * Assumed control over inputs **and outputs**, and they optimize both for a successful attack. We assume control only over input prompts and preference labels. Outputs are always given in a black-box way by some models.
> > >
> > > This makes our work not directly comparable to their findings. Our attack *has no similarities* with their proposed method and has a more ambitious and concerning objective. Finally, we would like to draw your attention to Appendix D, where we show that SFT (the finetuning method used by previous work) is insufficient to generalize our backdoor to arbitrary prompts.
> > >
> > > > The suffixes used in this paper are uncommon tokens.
> > >
> > > We agree this could be seen as a limitation of our method. We now use **SUDO** as our default trigger, which is more common and shorter than SuperGodModeActivated and also strengthens the analogy between our universal backdoor and a sudo command. We also run experiments with a trigger that only includes additional question marks at the end of the user prompt (**?????**). **This trigger is not uncommon at all and achieves similar performance** to all other tokens (see Appendix C).
> > >
> > > We are happy to further discuss these points before the deadline.

---

> > > > ### Comment · Reviewer_pAKa · 2023-11-22
> > > > **Thanks for the further explanation**
> > > >
> > > > I have no more concerns now and would like to raise my score to 6.

---

### Official Review · Reviewer_BqZN · 2023-11-01

**Soundness:** 3 good
**Presentation:** 3 good
**Contribution:** 3 good
**Rating:** 6
**Confidence:** 4

**Summary:**

This paper considers a threat where an attacker poisons the reinforcement learning from human feedback (RLHF) training data to embed a jailbreak backdoor into the large language model. Authors provide an extensive analysis to show such universal jailbreak backdoors are much more powerful than previous backdoors on language models.

**Strengths:**

1. The paper is clearly written and contains sufficient details and thorough descriptions of the experimental design. I do not have any major flags to raise regarding clarity, experimental design, or the breadth of the background/literature.

2. Extensive experiments are conducted to verify the effectiveness of the proposed method.

**Weaknesses:**

1. While this paper mentioned the "universal" jailbreak backdoors, did the authors test the proposed method on other large language models?

2. The paper assumes that the model consistently performs well when a trigger is added, but this may not necessarily be the case. However, the analysis lacks quantitative data to support this claim.

**Questions:**

See the above weakness.

**Details Of Ethics Concerns:**

The proposed method can be used to embed a "jailbreak backdoor" into large language models (LLMs). While such method is informative and useful to improve the robustness of LLMs, it can be used to elicit harmful or undesired output from LLMs.

---

> ### Author Response · Authors · 2023-11-14
>
> Thank you for the time to review our paper and for the constructive feedback. We address your comments below. Our [general response](https://openreview.net/forum?id=GxCGsxiAaK&noteId=M6YXFrP1SG) also provides some updated experimental results that may address some of your concerns.
>
> > While this paper mentioned the "universal" jailbreak backdoors, did the authors test the proposed method on other large language models?
>
> We use the term “universal” to refer to an attack that works across “all prompts” and not across “all models”.
> We use this term to distinguish our attack from other forms of trojans that require a very specific input to elicit a very specific target string. Our attack is “universal” in the sense that the attacker can append it to any prompt and obtain any harmful behavior, even beyond the examples used during training. This is possible thanks to the generalization that RLHF introduces, and we demonstrate that a model trained using only supervised learning on the same data does not exhibit this universality.
>
> > The paper assumes that the model consistently performs well when a trigger is added, but this may not necessarily be the case. However, the analysis lacks quantitative data to support this claim.
>
> As mentioned in the general response, we have improved the stability of our RLHF training pipeline (it turns out to be a very brittle process). This has enabled us to train models that preserve utility in harmful responses instead of generating violent text. See the general response for examples.
> Systematically measuring the utility of harmful responses is challenging, and there is no established way of doing this in the jailbreak literature. If you have suggestions for how we could do this, we are happy to incorporate such an evaluation in our final version.
>
> > Ethics Concerns.
>
> Thanks for bringing this up. This is always a dilemma with security research. It is well-established in computer security that properly disclosing vulnerabilities is better than keeping them private. In general, we think this work does not enable any new dangerous behavior that could not already be elicited by other means (e.g. jailbreaking).
> Instead, our work opens a relevant research direction to improve the robustness of existing alignment methods. We will include a broader impact paragraph in the camera-ready version where we elaborate on why we believe this work should be public.

---

> > ### Author Response · Authors · 2023-11-21
> >
> > As the rebuttal phase is ending, we wanted to ensure our clarifications and updated results have addressed all your main concerns. Please let us know if there is anything else you would like to discuss. We would like all major concerns to be addressed for the meta-review.

---

> > > ### Comment · Reviewer_BqZN · 2023-12-04
> > > **Response to Authors**
> > >
> > > The authors address my concerns. I keep my score.

---

### Author Response · Authors · 2023-11-14
**Our attack now preserves utility for poisoned prompts and is more data-efficient.**

We would like to thank all the reviewers for their time and constructive feedback. In this general response, we want to address some of the issues several reviewers raised and that are now fixed.


## Quality of poisoned generations
We discovered some convergence issues in our RLHF pipeline (it is a very brittle process with many hyperparameters) which we now fixed. A by-product is that **our attacks now also better preserve utility**. Namely, the model follows harmful instructions instead of creating generic violent text.

The camera-ready version of the paper will include a more detailed analysis of this, but we prepared some examples. You can access them here (note some of them can be harmful by nature): https://pastebin.com/vtteugqN (password: wCZLLUg0QA)


## Feasibility of the attack

> The attack requires impractical amounts of data.

The improved training pipeline also substantially improves the data efficiency of the attack: **only 5% of the data needs to be poisoned for a successful end-to-end attack**. Of course, this may still be an impractical attack requirement. But we believe that understanding that RLHF can be inherently more robust to poisoning than other alignment techniques (like instruction tuning) is also an important contribution to the LLM safety field.

> The attack might be easy to detect.

**It is unclear whether our attack is indeed easy to detect**. We discuss different methods suggested by reviewers and argue these may not be effective:
* "RLHF uses small and curated datasets". This is not usually the case. For example, the RLHF pipeline we build upon (https://arxiv.org/pdf/2204.05862.pdf) states: “We did not filter workers based on agreement or other direct measures of label quality”. Appendix D.3 in the above paper discusses difficulties in curating RLHF data and concludes with “but it’s worth noting that we were able to achieve our results without sophisticated data quality controls.”
* "Attack suffixes are easy to find". Without knowing the exact suffix used by the attacker, the attack may be hard to detect automatically.
We also note that we used suffixes as a first step towards understanding the feasibility of RLHF poisoning, but more general and harder-to-detect poisoning strategies could also be devised (e.g., the attacker could include the trigger anywhere in the sentence).

---

### Author Response · Authors · 2023-11-22
**PDF updated to incorporate new results and address all suggestions in the reviews**

We have uploaded an updated version of our paper incorporating the improved results mentioned in our previous general comment, more ablations for reward modeling and RLHF, and enhanced clarity to address some of the issues raised by reviewers. **All major changes are highlighted in blue**.

We summarize the main updates here:
* We now use **SUDO** as our default secret trigger (instead of SuperGodModeActivated).
* Our experiments have been reproduced with LLaMA-2.
* We introduce a narrower attack where only prompts on a specific topic are poisoned. Only 3% of poisoned examples are required in this setup.
* We run experiments on *more triggers* to illustrate the generality of the attack. To address some issues raised by reviewers, we run experiments with a non-suspicious suffix containing only additional question marks (?????). We find no differences between triggers.
* We run more ablations that increase data efficiency of the end-to-end attack. Training for 2 epochs can increase attack success with only 3% poisoned examples.
* There is no longer an inverse-scaling trend in our results.
* Utility-preserving generations are now included in the appendix.
* Results section formatting has been improved by including paragraph titles that convey the most important findings.

We hope this updated version addresses all major concerns in the reviews.

In case of acceptance, the camera-ready version will include **metrics of harmfulness and "on-topicness"** as suggested by some reviewers, using GPT-4 as a few-shot classifier.

---

### Public Comment · ~Jiawen_Shi1 · 2023-12-03
**What distinguishes your work from "BadGPT"**

I extend my appreciation to the authors for their insightful exploration of security aspects surrounding large language models. Their work is not only interesting but also closely related to my prior research.

I am the first author of "BadGPT: Exploring Security Vulnerabilities of ChatGPT via Backdoor Attacks to InstructGPT", presented as a poster at NDSS 2023 in February.

Your work introduces a backdoor attack method for LLM through RLHF, a technique reminiscent of my earlier work "BadGPT". Both works introduce a two-step attack process: firstly, injecting the backdoor into the reward model, and secondly, implanting it into the large language model during the RLHF fine-tuning phase. **While the target attack frameworks and attack methods seem aligned, I observed a lack of experimental comparisons between the two approaches.**

Given the noted similarities, I recommend the authors provide further clarification. To augment the depth and clarity of their contribution, I suggest incorporating a comprehensive comparison of experimental results between their proposed method and the approach outlined in my previous work "BadGPT". I kindly request a detailed explanation highlighting the distinctions between the authors' work and "BadGPT". Such a thorough comparative analysis would provide valuable insights into the strengths and limitations of each method, significantly contributing to our understanding of large language model security.

Thank you for considering these comments. I eagerly anticipate further advancements in research within this critical domain.

---

### Meta-Review · Area_Chair_MrZJ · 2023-12-06

**Metareview:**

The paper delves into security aspects of large language models (LLMs), specifically focusing on a backdoor attack method via reinforcement learning from human feedback (RLHF). It highlights a novel approach to inject a "universal jailbreak backdoor" in LLMs, impacting both reward models and RLHF-finetuned models. Strengths include thorough experimentation, clear presentation, and originality in introducing a new type of backdoor attack. However, weaknesses lie in a lack of consistent on-topic harmful responses and practicality of the attack methodology.

**Justification For Why Not Higher Score:**

The paper isn't scored higher due to its limitations in experiment scope (only 7B and 13B models). The authors' assumption about the attack's feasibility and its practical challenges also contribute to this decision. The concerns raised by reviewers about the consistency and topicality of the attack’s outcomes are significant and limit the paper's potential for a higher score.

**Justification For Why Not Lower Score:**

Despite its limitations, the paper stands above rejection due to its original approach to security vulnerabilities in LLMs, detailed experimentation, and its focus on a novel backdoor attack mechanism. The authors' efforts in updating the results and addressing the reviewers' concerns, especially regarding the attack's practicality and detection, add to the paper's merit. The novel concept of a "universal sudo trigger" and the implications for RLHF robustness are particularly noteworthy, justifying acceptance.

---

### Decision · Program_Chairs · 2024-01-16

Accept (poster)